# Compositional Discrete Latent Code for High Fidelity, Productive Diffusion Models

**Samuel Lavoie**[*]  **Michael Noukhovitch  Aaron Courville**
Mila, Université de Montréal

## Abstract

We argue that diffusion models' success in modeling complex distributions is, for the most part, coming from their input conditioning. This paper investigates the representation used to condition diffusion models from the perspective that ideal representations should improve sample fidelity, be easy to generate, and be compositional to allow out-of-training samples generation. We introduce Discrete Latent Code (DLC), an image representation derived from Simplicial Embeddings trained with a self-supervised learning objective. DLCs are sequences of discrete tokens, as opposed to the standard continuous image embeddings. They are easy to generate and their compositionality enables sampling of novel images beyond the training distribution. Diffusion models trained with DLCs have improved generation fidelity, establishing a new state-of-the-art for unconditional image generation on ImageNet. Additionally, we show that composing DLCs allows the image generator to produce out-of-distribution samples that coherently combine the semantics of images in diverse ways. Finally, we showcase how DLCs can enable text-to-image generation by leveraging large-scale pretrained language models. We efficiently finetune a text diffusion language model to generate DLCs that produce novel samples outside of the image generator training distribution. Code available: https://github.com/lavoiems/DiscreteLatentCode

## 1 Introduction

Denoising diffusion models [Ho et al., 2020] have demonstrated incredible capabilities for generating high-fidelity images [Peebles and Xie, 2023]. To achieve this feat, state-of-the-art methods [Ramesh et al., 2022, Rombach et al., 2022] generally train by conditioning on image labels or text captions [Radford et al., 2021]. Yet, strong diffusion models still generate images that exhibit low diversity or don't realistically reflect complex input prompts [Astolfi et al., 2024]. We posit that these failures of diffusion models are rooted in their inability to fully model the data distribution and can be alleviated by conditioning diffusion models on a better representation of the data. We argue that an ideal representation should (1) lead to *high-fidelity* generation of the data and (2) be *compositional* [Fodor and Pylyshyn, 1988, Hadley, 1997] to enable a generative model to *produce* novel images outside the training distribution by recomposing parts of images seen during training; i.e. enables *productive* generation [Edelman and Intrator, 2000] .

A common choice for conditioning diffusion models is with a text prompt or caption. Natural language is a flexible representation of the world [Whorf, 1956, Wittgenstein, 1953, Niu et al., 2024] that is easy to transmit and learn [Tomasello, 1999, Smith and Kirby, 2008], and is compositional [Johnson, 2004]; producing novel meanings by composing known words in new ways. However, text captions are poor descriptors of images [Foucault, 1966, Lavoie et al., 2024] as they only capture a few concepts of an image while excluding everything else e.g. background subject, contextual items, quality/resolution of the image itself. Consequently, diffusion models conditioned directly on a text

---

[*]Correspondence to samuel.lavoie.m@gmail.com.

39th Conference on Neural Information Processing Systems (NeurIPS 2025).

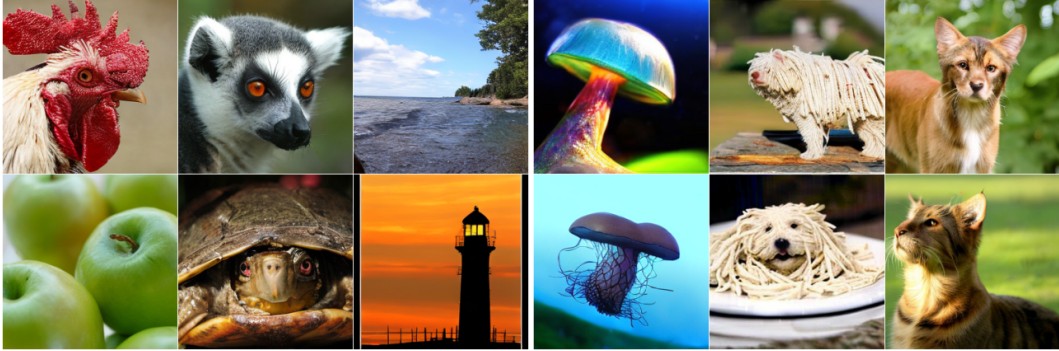

(a) Unconditional generation.      (b) Semantic compositional generation.

Figure 1: **Selected samples generated from a DiT-XL/2 with DLC$_{512}$ for both in-distribution and out-of-distribution (OOD)**. Model trained on ImageNet $256 \times 256$ conditioned on a Discrete Latent Code of $512$ tokens. **Left:** Samples from unconditional generation. **Right:** OOD samples of semantic compositional generation by conditioning on diverse compositions of two DLCs corresponding to (1) jellyfish and mushroom, (2) komodor and carbonara and (3) tabby cat and golden retriever.

representation [Radford et al., 2021], such as Stable Diffusion [Esser et al., 2024], often struggle to produce images consistent with the prompt [Huang et al., 2025]. Though modern text-to-image models have leveraged text to create intricate, novel images [Ramesh et al., 2022], they often miss the desired semantics of an image (e.g. ignoring word order [Yuksekgonul et al., 2023]). One solution is to enhance the captions to include more information about the image [Urbanek et al., 2024], but this is an expensive labelling task with humans and induces hallucinations when done with another model [Liu et al., 2024].

An alternative is to condition the generative model on a learned *image* representation. Specifically, image embeddings trained with self-supervised learning (SSL) [Hjelm et al., 2019, Oquab et al., 2024] are structured and more expressive than captions. But the standard embedding is continuous, and has two issues: (1) it is *difficult to learn* a continuous distribution in order to sample from it (Section 3), and (2) they are generally *not flexibly composable* and cannot combine the semantics of two images in diverse ways (Subsection 5.2).

We propose to condition diffusion models with a representation that combines the benefits of both image and text representation. We condition a generative model with **Discrete Latent Code** (DLC), a sequence of *discrete* image tokens. DLCs are derived from Simplicial Embeddings (SEMs) [Lavoie et al., 2022] that are a sequence of distributions over a vocabulary of image tokens learned with an SSL method. We show that DLCs improve data modeling and are easy to learn, achieving the state-of-the-art FID for unconditional generative modeling on ImageNet (example shown in Figure 1a). By leveraging a discrete representation extracted from a SSL representation, DLCs are compositional, such that DLCs can be composed and conditioned on by a diffusion model to produce diverse novel OOD images as composition of image features (shown in Figure 1b). Finally, we connect DLCs to large-scale pretrained language models to create a text-to-DLC-to-image pipeline. We show that DLCs conditionally generated from text prompt can be used to generate images outside of the image generative model's training distribution.

## 2 Background

**Continuous diffusion model.** Throughout this paper, we denote the observation data as $x \sim p_{\text{data}}(\mathcal{X})$. Denoising diffusion probabilistic models (DDPM) [Sohl-Dickstein et al., 2015, Ho et al., 2020] learns to iteratively reverse a pre-defined stochastic noising process that transforms data into an unstructured noise distribution over time. Typically, this process is defined via a Gaussian transition kernel that gradually perturbs an input sample by following a schedule over $T$ timesteps in [0, 1]. Assume that $p_0(\mathcal{X}) = p_{\text{data}}(\mathcal{X})$ is our data distribution and $p_1 = \mathcal{N}(\mathbf{0}, \mathbf{I})$ is an isotropic Gaussian. DDPM defines the forward noising process as follows:

$$x_t = \alpha_t x + \sigma_t \epsilon, \quad x \sim p_{\text{data}}(\mathcal{X}), \epsilon \sim p_1, \tag{1}$$

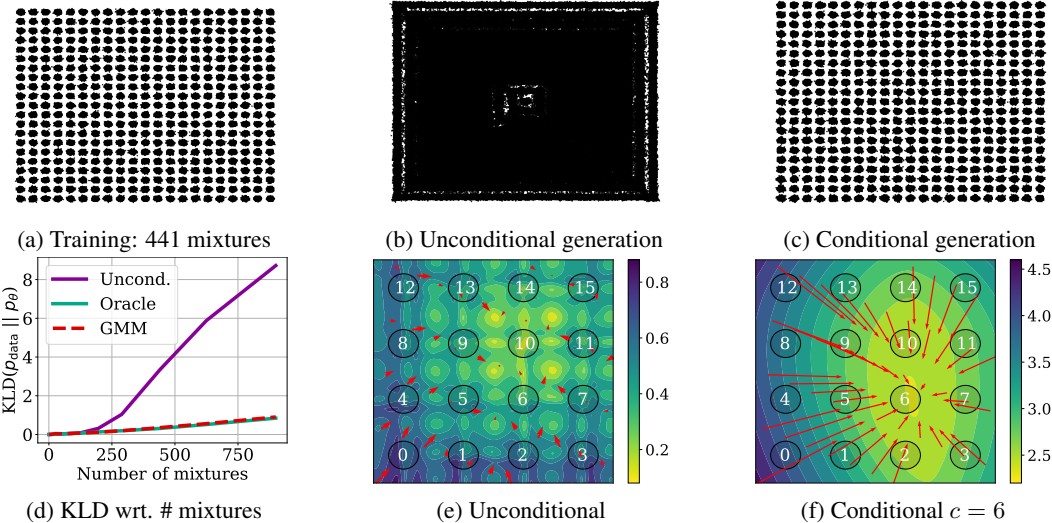

(a) Training: 441 mixtures     (b) Unconditional generation     (c) Conditional generation

(d) KLD wrt. # mixtures     (e) Unconditional     (f) Conditional $c = 6$

Figure 2: **Unconditional diffusion gets worse at fitting a distribution as the number of modes increases.** (a) Samples from the training data distribution $p_{\text{data}}$ with 121 mixtures. (b) Samples from unconditional diffusion model trained with $p_{\text{data}}$. (c) Samples from conditional diffusion model trained trained with $p_{\text{data}}$ and the ground-truth mixture index. (d) KL divergence between $p_{\text{data}}$ and the modeled distribution $p_\theta$ as we increase the number of mixtures. Unconditional's fit of the distribution degrades as the number of modes increase. Generations conditioned on an *oracle* index representing the mixture centroid and generation conditioned on an index inferred from a Gaussian Mixture Model (GMM) have good fit to highly modal distributions. (e, f) Heatmaps of the magnitude of the estimated score $s_\theta$ and vector fields with respect to the coordinate for the unconditional and the conditional generative models respectively. For d), we condition the score network on mixture index $c = 6$.

where $\alpha_t$ and $\sigma_t$ are defined according to a noise schedule (e.g. [Nichol and Dhariwal, 2021]).

Continuous diffusion models train a parameterized score network $s_\theta(\boldsymbol{x}_t, t, \boldsymbol{c})$ whose input is the noise sample $\boldsymbol{x}_t$, the timestep $t$ and an optional conditioning input (e.g. a label, a vector, a discrete code) $\boldsymbol{c}$ related to the input sample. The score networks are trained to estimate the noise vector $\boldsymbol{\epsilon}$ with the denoising score matching objective [Hyvärinen, 2005, Vincent, 2011] at all time steps. The DDPM objective is defined as a mean squared error loss between the estimated noise vector at a time step $t$ and the actual noise vector:

$$L(\theta) := \mathbb{E}_{t,(\boldsymbol{x},\boldsymbol{c}),\boldsymbol{\epsilon}} ||s_\theta(\boldsymbol{x}_t, t, \boldsymbol{c}) - \boldsymbol{\epsilon}||^2. \tag{2}$$

The score network may be used to define the reverse process which transforms a sample from the prior distribution $p_1$ into a sample from the data distribution using SDE solvers.

## 3 Generating highly modal continuous distributions is hard

Diffusion generative models work remarkably well on datasets with low diversity [Ho et al., 2020] such as LSUN [Yu et al., 2016] or when conditioned on a label [Peebles and Xie, 2023], or a text caption [Ramesh et al., 2021, Rombach et al., 2022]. However, large-scale diffusion models struggle to fit datasets with high diversity such as ImageNet [Li et al., 2024a], without conditioning. In this section, we demonstrate that this issue can be replicated with a simple toy dataset. We show that unconditional diffusion models exhibit a degenerate fit to the data distribution as the number of modes in the distribution increases. More precisely, given $p_{\text{data}}$ a ground truth training distribution and $p_\theta$ the model distribution. For a fixed model capacity and compute budget, the fit of $p_\theta$ to $p_{\text{data}}$ deteriorates as the number of modes in $p_{\text{data}}$ increases.

**Dataset and model.** Our data is a mixture of $N$ Gaussians on a square grid. For all $N$, we keep the variance fixed at $0.1$ and the distance between the center of two neighbouring Gaussians fixed at

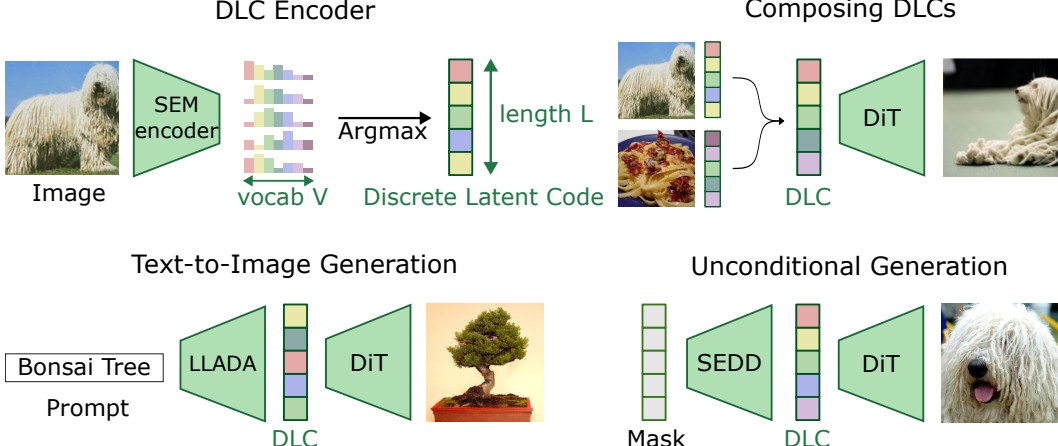

Figure 3: **Discrete Latent Codes (DLCs)** are **Top Left:** the output of a finetuned DINOv2 with SEM, followed by an argmax over the vocabulary. **Top Right:** we can generate semantically compositional images from a composition of two DLCs by selecting tokens from either code. **Bottom Left:** we enable text-to-image generation by finetuning a text diffusion model for text-to-DLC sampling. **Bottom Right:** we sample unconditionally by first sampling a DLC with SEDD then conditionally sampling an image with DiT.

1.0. [2] Our diffusion models are 4-layers MLPs trained using the improved DDPM procedure [Nichol and Dhariwal, 2021] with a batch size of 2048 and trained for 300M samples.

**Quantifying the degrading fit.** We provide an example of the training data $p_{\text{data}}$ for $N = 400$ in Figure 2a and observe a very poor fit to the data distribution from unconditional generation as shown in Figure 2b. Meanwhile, conditional generative models have a good fit to the training data as show in Figure 2c. We quantify the fit using the KL Divergence (KLD) between the ground truth distribution and the modeled distribution in Figure 2d. We estimate the density of the modeled distribution using the Gaussian Kernel Density estimation with bandwidth estimated using the Silverman rule. We find that the fit of the unconditional diffusion model quickly degrades (KLD increases) with the number of mixtures. Yet the same diffusion model trained with the same setup but conditioned on an *oracle* index representing the centroid of the mixture leads to a good fit for highly modal distribution. Conditioning on a *learned* index inferred via a Gaussian Mixture Model (GMM) is nearly identically good. We demonstrate in Figure 2f that the vector fields of conditional diffusion are simpler, linearly directing toward the mode. Comparatively, unconditional diffusion model learn a more complex flow as depicted in Figure 2e.

**Discussion.** These results illustrate how training unconditional diffusion models is harder on diverse, continuous datasets, and that conditioning on a representation, is an effective solution. Modeling continuous distributions with a large number of modes is difficult for diffusion models as it requires a more complex generative process than conditional generative models. Generating high-fidelity samples could be possible if we could represent the data to make it easier to learn and sample. In the next section, we show that images can be represented as a sequence of discrete tokens and that such representation is easy to learn and provide a good conditioning to image generative models.

## 4 Generative models with Discrete Latent Code

**Related work.** While earlier approaches on conditioning diffusion model primarily relied on labels or text captions [Dhariwal and Nichol, 2021, Rombach et al., 2022], recent works have explored conditioning on image embeddings [Preechakul et al., 2022, Bordes et al., 2022, Harvey and Wood, 2023, Pernias et al., 2023]. Like DLC, some works have conditioned generative models on discrete latent codes of an image [Lavoie-Marchildon et al., 2020, Bao et al., 2022, Hu et al., 2023, Wang et al., 2023, Xu et al., 2024] but they are generally using short codes of small dimensionality. In contrast,

---

[2]For sufficiently small variances or distances between Gaussians, we found that diffusion models struggle to capture the fine detail extant in the data [Karras et al., 2024].

we represent images as a longer, high-dimensional codes and leverage state-of-the-art SSL objectives for learning the image codes. This enables the code to represent more fine-grained features than prior work and to generally improve on image modelling. Several approaches have been proposed for learning discrete encoding [Oord et al., 2018] for image generation [Esser et al., 2021, Chang et al., 2022, Yu et al., 2022, Xu et al., 2024]. However, these approaches require back-propagating through the hard-discretization, necessitating a gradient estimator. Instead, we learn the discrete code using SEMs [Lavoie et al., 2022] which does not necessitate a gradient estimator. Recently, Li et al. [2024a] improved unconditional generative models by conditioning them on a continuous latent image representation. Their approach projects embeddings via a randomly initialized and frozen network, which makes them easily learnable but not compositional. Tangentially, REPA [Yu et al., 2025] also leverages DINOv2 but improves label-conditioned diffusion by aligning its latent representation with DINOv2.

**Inferring Discrete Latent Codes.** We leverage a SEM encoder [Lavoie et al., 2022] trained via a distillation objective [Zhou et al., 2022, Oquab et al., 2024]. Let $e_\theta(\boldsymbol{x}) \in \mathbb{R}^d$ an encoded representation. Each simplicial embedding $S_i = \sigma_\tau(e_\theta(\boldsymbol{x}) \cdot W_i)$ is a projection of the encoded representation onto the V-dimensional simplex, with a learnable linear projection $W_i \in \mathbb{R}^{d \times V}$ followed by a temperature-scaled softmax $\sigma_\tau$. Given a sequence of SEMs $(S_1, S_2, ..., S_L)$, we infer a discrete latent code $\boldsymbol{c}$, defined, by taking the argmax of each SEM. The DLC is thus defined as: $T_i = \arg\max S_i, i \in [L]$, $\boldsymbol{c} = (T_1, T_2, ..., T_L)$ where $T_i$ is a token that takes a value in $\mathbb{N}^V$. We show an overview in Figure 3.

**Improving unconditional generation with DLC.** As discussed in Section 3, learning a generative model on $p(\boldsymbol{x})$ may be hard, specifically in cases where $p(\boldsymbol{x})$ is highly modal. This work proposes to model $p(\boldsymbol{x})$ as the product of two generative models that are easier to learn. Specifically,

$$\underbrace{p(\boldsymbol{x})}_{hard} = \sum_{\boldsymbol{c}} \underbrace{p(\boldsymbol{x}|\boldsymbol{c})}_{easy} \cdot \underbrace{p(\boldsymbol{c})}_{easy} \tag{3}$$

Practically, sampling from $p(\boldsymbol{x})$ can be achieved by ancestral sampling. First, sample from $p(\boldsymbol{c})$. Conditioned on the sampled code $\boldsymbol{c}$, sample the image $p(\boldsymbol{x}|\boldsymbol{c})$.

**DLC conditioned diffusion models.** Given an image $\boldsymbol{x}$ and its associated discrete latent code $\boldsymbol{c}$, we train a conditional denoising score matching network $s_\theta$ using Equation 2 to model $p(\boldsymbol{x}|\boldsymbol{c})$. As discussed in Section 3, $p(\boldsymbol{x}|\boldsymbol{c})$ can be easy to model when $\boldsymbol{c}$ facilitates the conditional generation; i.e. is an expressive and well structured representation of $\boldsymbol{x}$. For image generation, we argue that such a representation of an image, can be obtained from a SOTA SSL encoder.

**Unconditional generation of DLC.** For Equation 3 to hold true, $p(\boldsymbol{c})$ has to be easy to model. Highly modal continuous distributions are hard to model with diffusion models as discussed in Section 3. Thus, a continuous code extracted from an SSL encoding of the image may also be equivalently hard to model. In contrast, large language models have shown the ability to model internet scale natural language, which are discrete and compositional codes representing highly diverse semantics. Thus, we argue and show that a discrete and compositional representation, such as a DLC, of a diverse dataset of images $p(\boldsymbol{c})$ is easy to generate too. Given that DLC are non-autoregressive, we propose to model $p(\boldsymbol{c})$ with a discrete diffusion model [Austin et al., 2021], specifically SEDD-Absorb [Lou et al., 2024]. SEDD-Absorb samples a discrete code by iteratively unmasking a fully masked-sequence. The token to be unmasked is determined via a learned concrete score $s'_\theta : \mathcal{C} \times \mathbb{R} \to \mathbb{R}^V$ which estimates a diffusion matrix that controls the mass transition from the mask token to the DLC token. Thus, $s_\theta$ allows us to estimate the transition probability $p_{t-\Delta t|t}(c_{t-\Delta t}|c_t)$ and sample from $p(\boldsymbol{c})$ via the reverse diffusion process using e.g. the Tweedie $\tau$-leaping [Lou et al., 2024] simulation algorithm.

**Remasking DLC.** Recently, it has been shown that remasking tokens improves sampling of discrete diffusion models [Wang et al., 2025]. While remasking is not required for sampling DLC, we found that remasking improves the generation quality of images. Thus, we also introduce a remasking strategy for SEDD-Absorb. Given the approximated posterior $p_{t-\Delta t|t}(\boldsymbol{c}_{t-\Delta t}|\boldsymbol{c}_t)$. Let $\eta$, the probability of remasking a token. Following [Wang et al., 2025], we apply the remasking on unmasked token (i.e. $c_t^i \neq m$) in the interval $t \in [t_0, t_1]$ and $\sigma = \eta$ if $t \in [t_0, t_1]$ and $\sigma = 0$ otherwise. We define the posterior with re-masking:

$$\hat{p}_{t-\Delta t+\delta|t}^i(c_{t-\Delta t+\delta}^i|c_t^i) = \text{Cat}((1-\sigma)x_t^i + \sigma m) \text{ if } c_t^i \neq m \text{ else } p_{t-\Delta t+\delta|t}^i(c_{t-\Delta t}^i|c_t^i). \tag{4}$$

where $\delta = \sigma \cdot (1 - t)$ a correction term on the step to take into account the remasked tokens.

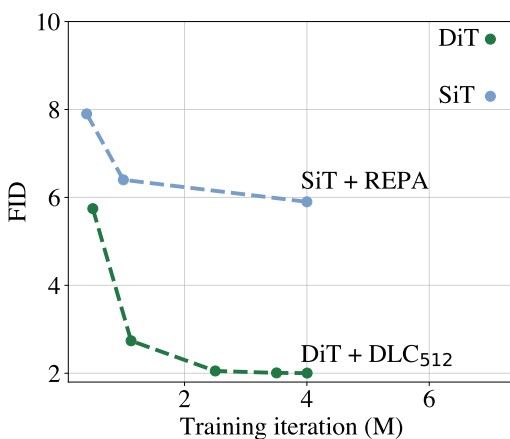

| | FID↓ | sFID↓ | IS↑ | PRE↑ | REC↑ |
|---|---|---|---|---|---|
| *Unconditional* | | | | | |
| RCG | 3.44 | - | 186.9 | - | - |
| **DLC$_{512}$** | **2.00** | **4.65** | **260.7** | **0.79** | 0.61 |
| *Conditional* | | | | | |
| MAR-H | 2.35 | - | 227.8 | 0.79 | 0.62 |
| SiT-XL/2 | 8.61 | 6.32 | 131.7 | 0.68 | **0.67** |
| REPA | 5.90 | - | - | - | - |
| DiT-XL/2 | 9.62 | 6.85 | 121.5 | 0.67 | **0.67** |
| *Uncond. w/ CFG* | | | | | |
| DisCo-Diff[†] | 3.70 | - | - | - | - |
| RCG | 2.15 | - | 253.4 | - | - |
| **DLC$_{512}$** | 1.59 | **4.16** | 255.4 | 0.81 | 0.63 |
| *Cond w/ CFG:* | | | | | |
| SiT-XL/2 | 2.07 | 4.49 | 277.5 | 0.83 | 0.59 |
| REPA | **1.42** | 4.70 | **305.7** | 0.80 | 0.65 |
| DiT-XL/2 | 2.27 | 4.60 | 278.2 | 0.83 | 0.57 |

Figure 4: **DLC greatly improves training efficiency for FID without CFG on ImageNet.** Evaluating FID w/o CFG during intermediate steps, DLC is already improving on vanilla DiT performance at 1/4 of the steps. Baseline numbers taken from Yu et al. [2025]

Table 1: **DLC achieves SOTA FID, sFID on unconditional ImageNet generation** while reducing reliance on CFG. Baseline numbers taken from published works. [†]: On ImageNet $64 \times 64$.

## 5 Image generation with DLCs

We investigate the impact of conditioning image diffusion models on Discrete Latent Code (DLC) inferred from a SEM encoder [Lavoie et al., 2022]. We demonstrate that diffusion models + DLC:

1. push the state-of-the-art on unconditional ImageNet generation (Table 1, Figure 4),

2. outperform generative model conditioned with continuous SSL embeddings (Table 2),

3. exhibit compositional generation (Figure 6),

4. exhibit increased image generation FID as we increase the sequence length (Figure 5).

**Setup for inferring DLC.** Following REPA [Yu et al., 2025], we leverage a pre-trained DINOv2 ViT-L encoder [Oquab et al., 2024] and a randomly initialized linear projection layer, projecting its output into SEMs. Additionally, we use a randomly initialized DINO head that takes the concatenated SEMs as input. We investigate three SEMs configurations with the same total number of dimension (131 072): (1) 32 tokens of dimension 4096 ($32 \times 4096$), (2) 128 tokens of dimension 1024 ($128 \times 1024$) and (3) 512 tokens of dimension 256 ($512 \times 256$), to systematically understand the trade-off between the number of tokens, the vocabulary size and their effective capacity. For the same number of tokens, larger sequence length can represent more combinations (i.e. $512^{256} > 1024^{128} > 4096^{32}$).

We use the DINOv2 codebase with minimal modifications to support SEM training and we re-use the same hyper-parameter used for DINOv2 pre-training. We train the SEM encoders on ImageNet-1K [Russakovsky et al., 2015] for 100 epochs. After pre-training, we associate each image with its DLC by taking the argmax across the SEMs outputs, as shown in Figure 3.

**Setup for training image diffusion.** Our image diffusion experiments build upon the DiT codebase [Peebles and Xie, 2023]. We use the DiT-XL/2 architecture for latent diffusion on VAE-encoded images [Rombach et al., 2022]. We minimally modify the code to replace class label conditioning with either DLCs or continuous self-supervised embeddings, both of which are pre-computed. DiTs are trained on ImageNet for up to 800 epochs, using the same optimizer, learning rate, global batch size, sampling strategy, and other hyperparameters as in the original DiT implementation. We embed the tokens of the discrete latent code with an embedding matrix, as commonly done for embedding the label [Dhariwal and Nichol, 2021]. These embedded tokens are averaged to form the conditioning input to the DiT [Peebles and Xie, 2023]. We add an additional *drop-token* to the embeddings to also train an unconditional model and to leverage classifier-free guidance [Ho and Salimans, 2022].

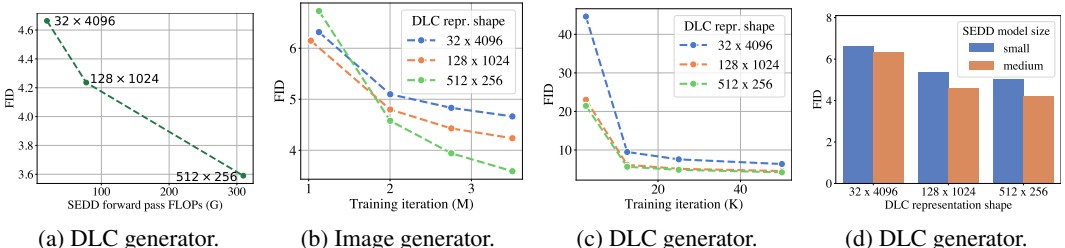

| (a) DLC generator. | (b) Image generator. | (c) DLC generator. | (d) DLC generator. |

Figure 5: **Scaling analysis of DLC: trade-off between performance and compute controlled via the sequence length.** (a) FID with respect to compute : FID and compute scale with the sequence length of DLE. (b) and (c) Training a generative model to generate long sequence length and training an image generative model conditioned on long sequence length converge to a lower FID. (d) Larger sequence length are more sensitive to the model size and attain lower FID. Results obtained without CFG nor remasking. Unless mentioned otherwise, DiTs are trained for 500 epochs.

**Setup for latent embeddings diffusion.** We use SEDD [Lou et al., 2024], a discrete diffusion model, to generate DLCs corresponding to ImageNet images. We closely follow their training recipe but instead of text tokens, our model is trained on DLC tokens. We accordingly adjust the sequence length and the vocabulary size of the model to match each DLC configuration. For example, the $512 \times 256$ DLC setup will be modeled by a network that outputs a sequence of size $512$ with a vocabulary size of $256$. We use the re-masking strategy during sampling, introduced in Section 4, in our main results. Additional analysis on the impact of remasking is in Appendix A. All other decision for training and sampling SEDD, including the model definition, training optimizer, training code, and hyperparameters remain unchanged from the original SEDD setup for text diffusion.

To compare with discrete, we also train a continuous diffusion model to generate DINOv2 embeddings. We strictly follow the procedure used to train RDM [Li et al., 2024a] which leverages the DDPM objective with a U-NET backbone. We train the diffusion model with the same depth as the medium SEDD model to ensure a fair comparison between continuous and discrete embeddings.

## 5.1 DLC improves sampling fidelity

**DLC improves unconditional generation.** DiT [Peebles and Xie, 2023] and SiT [Ma et al., 2024] are strong and widely used backbones for label-conditioning ImageNet generation. In contrast, unconditional generation of ImageNet typically yields lower quality samples, as measured by FID, which aligns with our observations in Section 3. ImageNet is a diverse dataset with lots of modes (e.g. different classes, environments, etc.) This feature of ImageNet hinders continuous diffusion models' ability to accurately fit the data distribution effectively. Prior works improved on unconditional generation by conditioning generative models on learned representations, but underperform compared to label conditioning [Li et al., 2024a, Xu et al., 2024]. In Table 1, we present a system-level comparison of the state-of-the-art conditional and unconditional generative models. We find that a DiT-XL/2 model with DLC considerably improves the FiD compared to the same DiT-XL/2 with label-conditioning. Additionally, DiT trained with DLC pushes the unconditional generation state-of-the-art with a FID of 1.59 and closes the gap with label-conditioned generative models.

**DLC pushes the SOTA for generation without guidance.** Figure 4 compares the performance of various diffusion methods training iterations without classifier-free guidance. We observe that DiT trained with DLC achives significantly lower FiD than label-conditioned baselines. Additionally, it achives a lower FiD than the baseline methods in considerably fewer training iterations. In Table 1, we report MAR-H [Li et al., 2024b], the current SOTA generative model without guidance. We find that DLC-conditioned DiT outperforms MAR-H with a FID of 2.00 achieving the SOTA ImageNet generation without any guidance including label conditional and unconditional generation.

**DLCs enable a trade-off between performance and compute.** The plots in Figure 5 depicts the trade-off between the sequence length and the vocabulary size in DLCs. Although all configurations are designed to have the same total dimensionality, short sequence length (i.e. fewer tokens but larger vocabulary) are cheaper to obtain but lead to worst performance compared to long sequence length as shown in the scaling plot in Figure 5a. Figure 5b and Figure 5c further show that both image

| | Linear probe % ↑ | Gen. FID↓ | Enc-Dec FID ↓ |
|---|---|---|---|
| Unconditional | - | 27.30 | - |
| RCG | 0.1* | 4.89 | - |
| DINOv2 | 85.9 | 37.90 | 2.12 |
| $DLC_{512}$ | 85.3 | 4.21 | 4.09 |

| | Vendi ↑ |
|---|---|
| RCG | $4.2 \pm 0.9$ |
| DINOv2 | $9.6 \pm 1.3$ |
| $DLC_{512}$ | $13.2 \pm 0.8$ |

Table 2: **DiT + DLC attains both high linear probe and low generative FID.** Comparing DiT-XL/2 trained for 400 epochs with no CFG nor remasking. Unconditional and RCG conditional generation numbers are taken from [Li et al., 2024a]. We evaluate the classification accuracy (%) of a linear probe trained on the input representation given to the diffusion model, the generative FID and the encoding-decoding FID obtained by encoding 50 000 ImageNet samples to condition the generative model. *Using the encoder of RCG's released model.

Table 3: **DLC produces more diverse generations**. Vendi score, measuring diversity of 8192 generated composed samples. Avg.±std across 10 randomly chosen pairs of image embeddings.

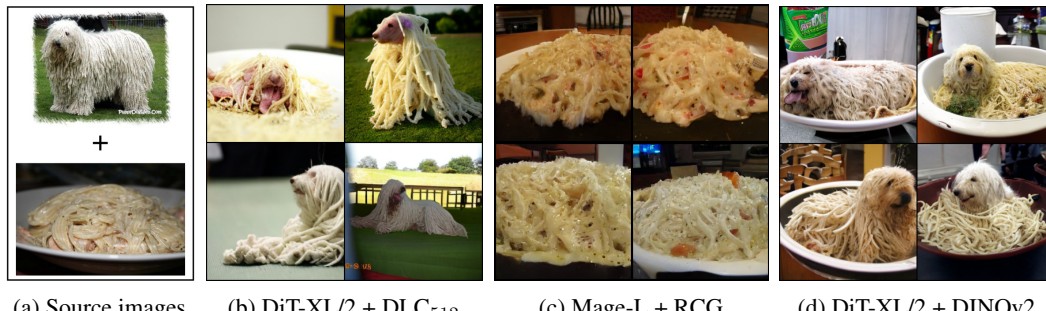

(a) Source images     (b) DiT-XL/2 + $DLC_{512}$     (c) Mage-L + RCG     (d) DiT-XL/2 + DINOv2

Figure 6: **DLC enables high quality, diverse generations for semantic composition** We generate compositions of (a) Komondor (dog breed) and Carbonara (b) DiT + DLC generates diverse images of a dog with pasta-like fur (c) Mage-L + RCG fails to generate a dog (d) DiT conditioned on the averaged DINOv2 embedding produces reasonable but not-diverse images.

and DLC diffusion models take more iterations to converge for longer sequence but ultimately reach a lower FID. Lastly, Figure 5d shows that models trained with longer sequences benefit more from larger DLC generators compared to those trained with smaller embeddings.

**DLC improves over continuous embeddings conditioning.** A natural alternative to DLC is to condition the generative model on continuous SSL embeddings. However, as discussed in Section 3, continuous distribution that lie on a complex, multi-modal manifold, are hard for diffusion models to faithfully learn. The distribution of a SSL embeddings may also have a have a lot of modes and thus also be hard to learn. RCG addresses this issue by projecting the SSL embedding using a randomly initialized MLP, and report strong unconditional generation FID. Such projection likely results in a latent space with a simpler distribution and thus easier to learn. But, it also leads to a representation without structure that is unable to produce novel OOD samples.

In Table 2, we quantitatively compare unconditional diffusion models. Consistent with Section 3, both unconditional DiT-XL/2 model and DINOv2-conditioned DiT-XL/2 model yield high generative FID. However, the DINOv2 conditioned generative model provided with ImageNet-encoded continuous embeddings, instead of generated, achieves significantly lower FID compared to all of the methods. This results suggests that the poor generative FID of DINOv2 conditioned generative model stems from the inability of correctly modeling the DINOv2 embeddings. Conversely, RCG which conditions its generative model on a SSL embedding projected through a randomly initialized network can attain a decent generative FID, likely because its randomized embedding distribution is easier to model. However, these embeddings lack semantic structure, as illustrated by the low linear probe accuracy of a classifier trained on the RCG embedding, which has negative consequences for its compositionality.

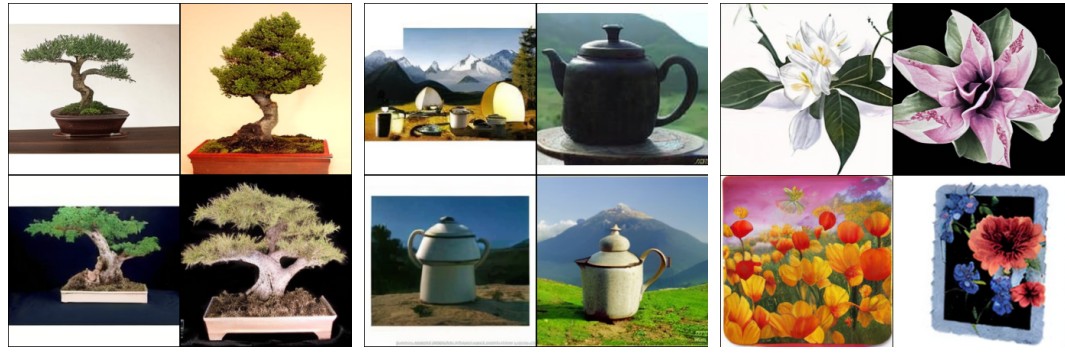

| (a) A bonsai. | (b) A teapot on a mountain. | (c) A painting of a flower. |

Figure 7: **DLC enables further OOD generalization with text-to-image generation.** We use a text-to-DLC model to enable generation of (a) bonsai, despite the label "bonsai" not being part of ImageNet, though there are a handful of images of bonsai. (b) a teapot on a mountain and (c) painting of a flower though neither exists in ImageNet's training set. We finetune LLaDA-8B to generate DLCs for a frozen DiT-XL/2 + $DLC_{512}$ trained on ImageNet-1k. The DLC tokens are generated by finetuning LLaDA [Nie et al., 2025], a text-conditional discrete diffusion model, on a random subset of 9M image-text pairs from LAION [Schuhmann et al., 2022].

## 5.2 DLC enables compositional diverse generation.

Compositional representations allows parts of representations to be composed in order to create novel concepts. A productive generative model can then leverage the composed representation to produce novel samples. In Figure 6, we explore the ability of diffusion models to produce novel images by composing the representations of two reference images from different classes (e.g. a komodor and carbonara). We compare models conditioned on a Discrete Latent Code with baseline continuous embeddings (RCG and DINOv2). We compose the continuous embeddings by averaging the embeddings of the reference samples. For DLCs, we construct a hybrid sequence by randomly sampling each token position from one of the two source DLCs, allowing for diverse combinations of semantic features. We show results in Figure 6a and observe that samples generated from composed RCG embeddings predominantly depict pasta, indicating a failure to represent both source concepts. DINOv2-based compositions do exhibit some degree of compositional generation, but the outputs show limited diversity and tend to recombine the same dominant features (e.g. dog in pasta). In contrast, DLC-based compositions successfully integrate visual features from both reference images and exhibit greater sample diversity (e.g. pasta on dog), as shown in Figure 6b. We quantify this diversity using the Vendi Score [Friedman and Dieng, 2023] in Table 3, and find that DLC compositions consistently outperform continuous embeddings in generating diverse and semantically blended samples.

## 6 Text-conditioned DLC for image generation

The dominant approach for conditioning image generative models is via a text prompt [Ramesh et al., 2022]. However, training these text-conditioned models typically necessitates hundred of million of image-text pairs [Schuhmann et al., 2022] and inaccessible compute for most practitioners. We show that these issues can be reduced by leveraging large-scale pretrained language models (LLM).

Discrete Latent Codes offer a promising alternative for text-conditioned image generation by directly leveraging LLMs. Since DLCs are discrete tokens, they can be treated as part of a language model's vocabulary and jointly be trained with text. Concretely, we propose to view the text-to-image generation as a Markov Chain: $text \rightarrow c \rightarrow x$. First, a text-to-DLC model samples a DLC from a text prompt using a language model, $p(c|text)$. Next, we generate an image from our DLC via a pre-trained image diffusion model $p(x|c)$. Together, we have a flexible text-to-image pipeline $p(x|text) = \sum_c p(x|c) \cdot p(c|text)$ that leverages pre-trained LLMs and pre-trained image generators.

To obtain a text-to-DLC generative model we build on top of LLADA [Nie et al., 2025], a pre-trained text diffusion language model with 8B parameters. We extend its vocabulary by adding $V + 1$ new

tokens corresponding to our DLC vocabulary and a `<START-DLC>` token. We learn to generate DLC's tokens conditioned on text by finetuning with masked-token prediction, as in pretraining LLADA. We train on image-caption pairs from LAOIN [Schuhmann et al., 2022] using a simple prompt of following the template `"<image-DLC><START-DLC><caption>"`. Notably, we show that we can obtain a text-to-image generative model with 9M image-caption pairs, orders of magnitude less than than the standard text-conditioning model.

We evaluate this pipeline's ability to generate novel images using text prompts out-of-distribution (OOD) relative to the image diffusion model's ImageNet training. As shown in Figure 7, we generate images with prompts that are clearly not ImageNet classes. To assess novelty, we retrieve the nearest example from ImageNet to each generated sample (in DINOv2 embedding space) and show them in Appendix D. Though some "bonsai" images exist in the dataset, they are not labelled as such, so our text-to-DLC demonstrates how we can leverage even unlabelled images. "A teapot on a mountain" shows a composition of two existing classes that are never seen together as we find no meaningful matches for teapots on a mountain in ImageNet. "A painting of a flower" demonstrates style transfer and decomposition as no paintings of flowers exist but they are painted or shown on other objects e.g. plates. Altogether, the fine-tuned LLADA has effectively learned to map text prompts to plausible DLCs—despite only having seen 9M image-caption pairs—and that the frozen diffusion model can interpret these DLCs to produce visually coherent and novel images outside of its training set.

Overall, this experiment shows that DLCs can serve as a compact and compositional bridge between LLMs and image diffusion models. Our pipeline is both effective and flexible, leveraging pretrained language and vision components without requiring joint training from scratch. By enabling LLMs to "speak in DLCs," we open up a new avenue for text-to-image generation that is scalable, modular, and generalizes beyond fixed label vocabularies.

# 7    Conclusion

We present Discrete Latent Code, a compositional discrete representation learned solely from images. DLC improves on the state-of-the-art for unconditional image generation on ImageNet, and enables productive generation capabilities leading to diverse compositional image synthesis and a novel text-to-image paradigm. Diffusion model research generally focuses on improving the diffusion process, assuming label or CLIP embeddings as a given. We believe that focusing on the structural properties of representations, such as compositionality, can enable greater progress in the field. Furthermore, discrete image embeddings are an underexplored direction but are becoming more relevant as large scale pretrained *diffusion* language models offer a vision for a unified text-image generation interface.

### Acknowledgments

This research is supported by Samsung, NSERC. The authors thank Oumar Kaba, Sébastien Lachapelle, David Dobre, Muqeeth and Paul Barde for the insightful discussions. MN is supported by the Fonds de recherche du Québec, Nature et Technologies. This research was enabled by compute resources and software provided by Mila (mila.quebec), Calcul Québec (www.calculquebec.ca), and the Digital Research Alliance of Canada (alliancecan.ca).

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

# A  Resampling DLC with SEDD-absorb

Resampling tokens in discrete diffusion models is akin to adding noise in DDPM sampling. Generally, in MCMC sampling, such noise may prevent the sampling chain from ending up into a local minima. In Equation 4, we introduced a remasking scheme for SEDD-absorb. In Figure 8, we explore the effect of the remasking ratio parameter $\eta$ on the ImageNet generation FID. We find a U-shape curve where a resampling ratio that is too low or too high cause degraded image generation quality.

In Figure 9 we present uncurated results with $\eta \in (0., 0.5)$. $\eta$ that are too low or too high produce samples that are lower quality, but for different reasons. $\eta$ close to 1 will result in a sampling with too little steps to produce good DLC (i.e. it is equivalent to having a very small total number of time-steps). $\eta$ that is too small will not allow the sampling of the DLC to correct mistakes made early in the sampling and will result in a local minima. In contrast to classifier-free guidance, we don't find that very high $\eta$ cause the model to generate weird artefacts or to reduce the diversity. This implies that remasking DLCs with SEDD-absorb or other remasking methods could be used as a strategy to wholly replace classifier-free guidance. That said, remasking and classifier-free guidance are compatible as using one strategy do not prevent from using the other strategy.

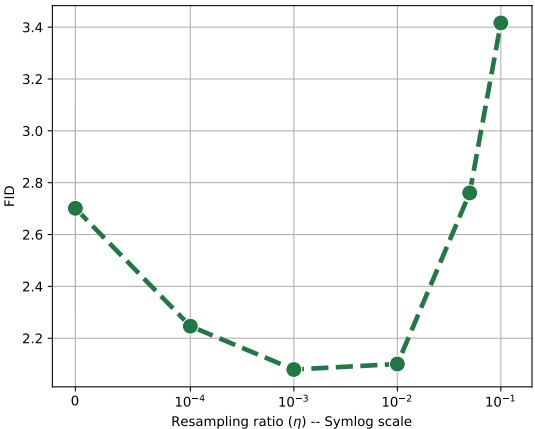

Figure 8: Effect of the resampling ratio in SEDD-absorb. Model trained on ImageNet $256 \times 256$ with $\text{DLC}_{512}$ a sequence of 512 DLC with 256 tokens each. We sample the tokens for 4096 steps and we activate the remasking for steps in [0.3, 0.55]. Generation without classifier-free guidance. We find a U-shape curve with an optimal resmapling ratio of $\eta = 0.01$.

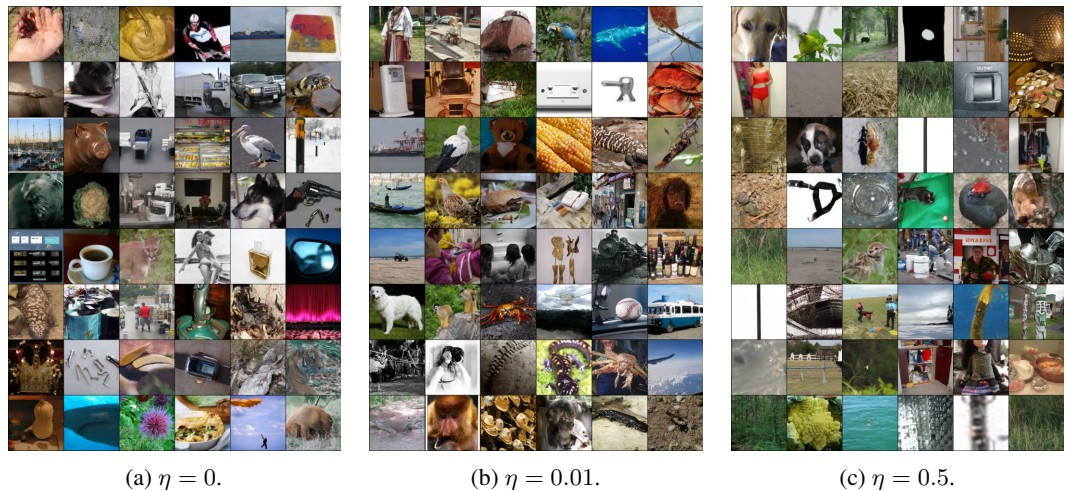

(a) $\eta = 0$.  (b) $\eta = 0.01$.  (c) $\eta = 0.5$.

Figure 9: Uncurated unsupervised generation for ressampling ratio $\eta \in (0, 0.01, 0.5)$. Generation without CFG.

# B  Comparing DLC to Stable Diffusion

To demonstrate the benefit of DLC, we compare to the state-of-the-art open-source diffusion model, Stable Diffusion, which is both larger and trained on more data. We aim to reproduce one of our productive examples, Carbonara + Komondor, from Figure 1b. First, we generate using only a text prompt "Komondor made of Carbonara". Next, we aim to compare as closely to our average-image-embedding conditioning. We leverage IP-Adapter [Ye et al., 2023], the standard tool for image-conditioned generation. We get the CLIP embeddings from our komondor and carbonara images and generate with IP-Adapter conditioning on the average of the two image embeddings. For all generations, we use IP-adapter's recommended version Stable Diffusion 1.5, we set IP-adapter scale to 0.6, CFG to default 7.5, and generate with 100 timesteps.

We show results in Figure 10. We find that text-only conditioning is not sufficient to generate our carbonara dog, supporting our claim that text embeddings can not always sufficiently capture image semantics. Generating from the average embedding is slightly better, though lacking in diversity and failing to generate a dog at all in one case. Finally, combining both text and image conditioning allows Stable Diffusion to approach our method, though it is clearly heavily leaning towards Komondor and only changed the fur to be more pasta-like.

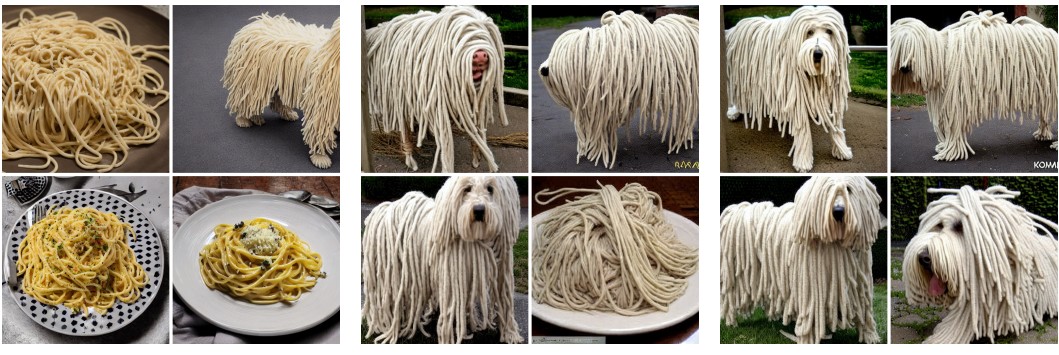

(a) Only text "Komondor made of Carbonara"  (b) Average of CLIP embeddings with IP-Adapter  (c) Average embedding and text

Figure 10: **Stable Diffusion can somewhat reproduce our combination of Komondor and Carbonara** but a) conditioning on purely a text prompt is insufficient and we require b) conditioning on the average CLIP embedding of an image of Komondor and Carbonara to achieve a reasonable combination of the classes, though one failure mode. Conditioning on c) both average image embedding and text prompt works best, though shows a distinct lack of diversity compared to our DLC-based method.

# C  Additional evidences showing inability of diffusion models to generate highly modal continuous distributions

Section 3 demonstrated that diffusion models struggle to model highly modal distributions. For completeness, Figure 11 showcases the full generations resulting from 9 mixtures, 121 mixtures, 400 mixtures and 900 mixtures for unconditional, oracle conditioned and GMM conditioned diffusion models. Unconditional generation observes a degrading fit as the number of modes in the dataset increase. Conditional generative model demonstrate a good fit even for very large number of modes. Interestingly, inferring the mixtures scales relatively well with the number of modes.

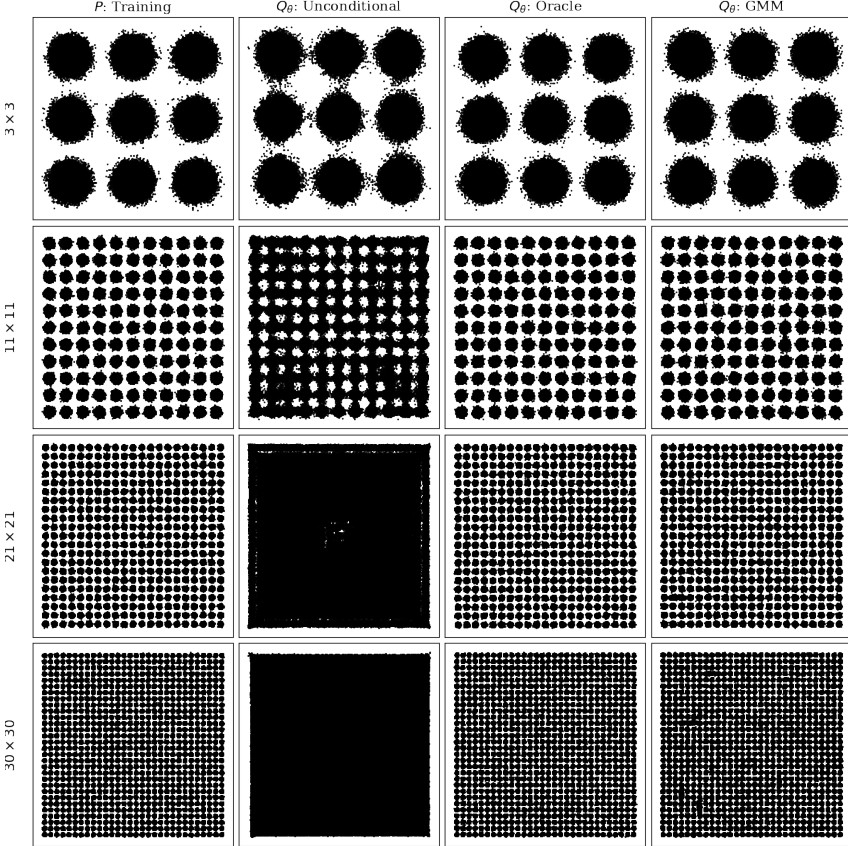

Figure 11: Comparing a $m \times m$ grid of mixture of Gaussian $P$ with samples from model distributions $Q_\theta$ unconditional, conditioned on an *oracle* mixture index and a mixture index inferred from a Gaussian Mixture Model (GMM). Unconditional generative models cannot samples highly modals distributions. Meanwhile, conditional generative models have no problem sampling highly modal distribution. Moreover, inferring mixture indices via a GMM also scales to highly modal distributions.

## D  Evaluating Novelty of Text-to-Image samples outside ImageNet

Our text-to-DLC-to-image pipeline enables generation of interesting images that reflects generalization outside of the ImageNet distribution. Evaluating the novelty of generations is a fundamentally difficult task [Kingma and Gao, 2023]. To give a qualitative sense of our pipeline's ability, we find the closest examples to our generation in the ImageNet training set. To do so, we use the DINOv2 embedding space and find the 5 nearest neighbours in that space. We show results in Figure 12.

We show three types of generalization found in our model. First, we show that this model can generate samples of images that occur in ImageNet, such as a bonsai, but for which no label of bonsai exists. Such generalization showcase the utility of open-ended generation. Second, we show compositional generalization where we generate teapot on a mountain. Notably, there are no images of teapots on mountains in the dataset. Our model clearly manages to learn and combine the semantics of separate foreground object and background setting. Thus, this results demonstrate that the generative image model can produce novel samples by extracting attributes in ImageNet and recomposing them in novel ways. Finally, we denote the generation of painting of flowers. While some painting of flower does exists, specific painting of the flower generated do not exists in the dataset. For example, there are not painting of the white flower in ImageNet.

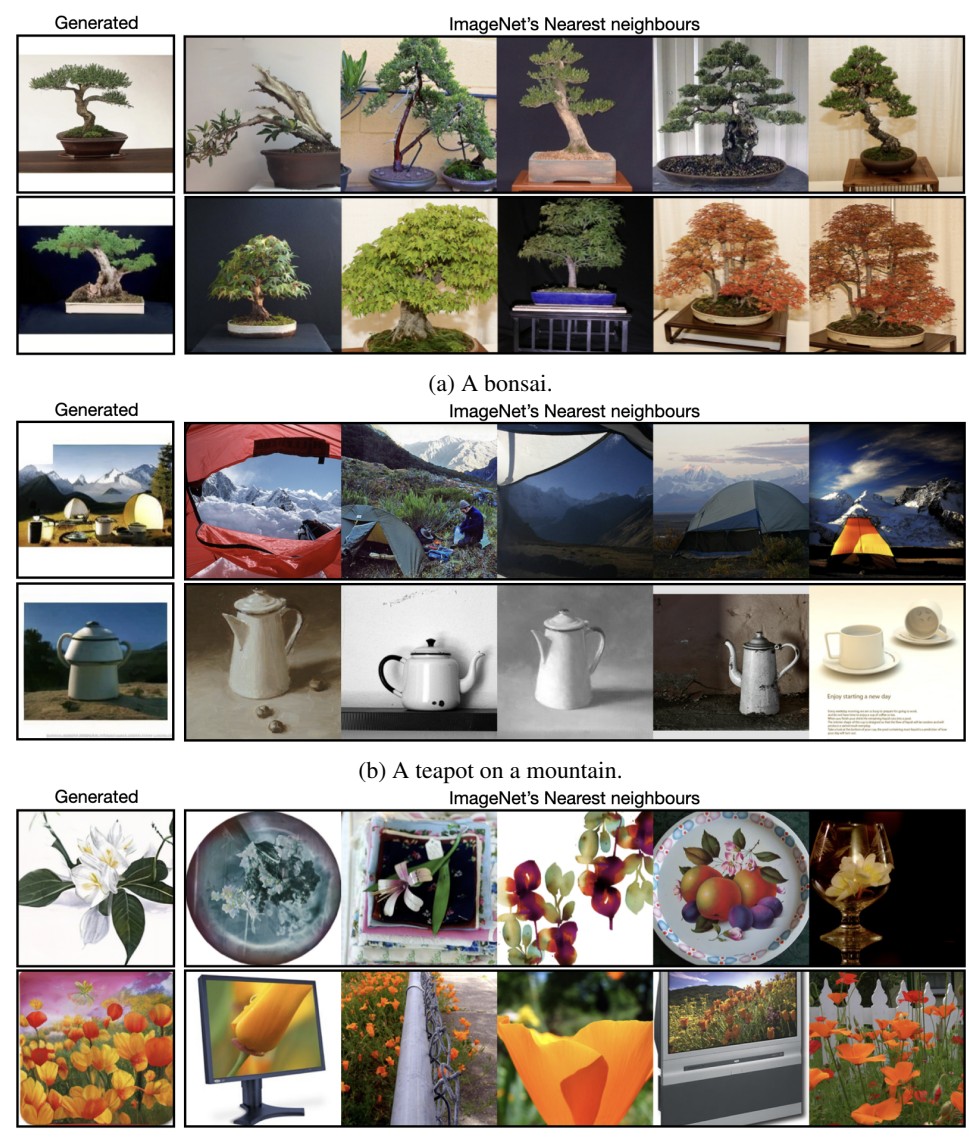

(a) A bonsai.

(b) A teapot on a mountain.

(c) Painting of a flower.

Figure 12: Text-to-image generated samples from Figure 7 and their semantic nearest neighbours in ImageNet's training set. We find the semantic nearest neighbours with respect to the cosine similarity of DINOv2-vit/l encoding. While bonsai is not part of the labels of ImageNet, we find some *bonsai* in ImageNet's training set that are similar to the generated samples. However, there are not *teapot on a mountain* nor *painting of a flower* that resemble those generated by our model. This result shows the productive capability of our image generative model.

# E    Experimentation details

**Unconditional image generation**    All models were trained with a batch size of 512 and trained according to the hyper-parameters specified in Table 5. For unconditional generation of SEMs, we use the Tweedie denoiser. We use remasking for generating the samples in Figure 4, Table 1 and Figure 8. Otherwise, no remasking is used. We generate 50000 SEMs that are then used to generate images using the image diffusion model.

The image diffusion models are all trained with a DiT-XL/2 model [Peebles and Xie, 2023] and the hyper-parameters specified in Table 4. Following Peebles and Xie [2023], we use the EMA VAE model from Stable-Diffusion. The generation uses DDPM [Ho et al., 2020]. We only use

| | | | | | |
|---|---|---|---|---|---|
| N blocks | 28 | | N blocks | 24 |
| Hidden size | 1152 | | Hidden size | 1024 |
| Patch size | 2 | | Num heads | 16 |
| Num heads | 16 | | Optimizer | AdamW |
| Optimizer | AdamW | | Learning rate | 3e-4 |
| Learning rate | 1e-4 | | Batch size | 512 |
| Batch size | 256 | | Warmup | 2500 |
| Weight decay | 0 | | Gradient clipping | 1.0 |
| Num sampling steps | 250 | | Weight decay | 0 |
| CFG scale (Table 1) | 1.4 | | Num sampling steps | 4096 |
| Training epochs (Table 1) | 1200 | | Resampling ratio $\eta$ (Table 1) | 1e-4 |
| Image size | 256×256 | | Training epochs (Table 1) | 200 |

Table 4: DiT-XL/2 hyper-parameters for training and sampling

Table 5: SEDD-medium hyper-parameters for training and sampling

| | |
|---|---|
| Model size | 8B-base |
| # sample seen | 9M |
| Optimizer | AdamW |
| Learning rate | 1e-5 |
| Total batch size | 128 |
| Grad. accum. steps | 8 |
| DLC shape | $128 \times 1024$ |

Table 6: LLADA text-and-DLC hyper-parameters for fine-tuning.

classifier-free guidance for reporting results in Table 1. For fair comparison, we re-use the same CFG scheme as DiT and apply CFG only on the first three-channel. For FID computation, we generate 50000 samples conditioning on the pre-sampled SEMs.

**Text-and-DLC fine-tuning** We fine-tune a LLADA-8B-base [Nie et al., 2025], a large diffusion language model parameterized as 8B parameters Llama [Grattafiori et al., 2024] transformer [Vaswani et al., 2023]. Contrary to SEDD, which predicts the concrete score, LLADA predicts the probability of every tokens directly $p_\theta(x_0^i|x_t)$. The training objective to train the transformer is the cross-entropy loss:

$$L(\theta) = -\mathbb{E}_{t,\boldsymbol{x}_0,\boldsymbol{x}_t}\left[\frac{1}{t}\sum_{i=1}^{L}\mathbf{1}[x_t^i = M]\log p_\theta(x_0^i|x_t)\right]. \tag{5}$$

For fine-tuning, we re-use the same Equation 5. However, instead of considering text tokens only in our objective, we consider pairs of text and DLC tokens. The DLC tokens comes from encoded images and the pairs text and images are randomly sampled from LAION [Schuhmann et al., 2022]. As a proof-of-concept, we randomly subsample 9M image-text pairs from LAION that are used for fine-tuning.

For sampling the DLC, we provide a masked sequence for 128 mask tokens followed by a separator token and the prompt. We follow Nie et al. [2025] protocol with low-confidence remasking strategy.

**Computational resources.** We fine-tune the DINO + SEM encoders on two A100 GPUs, with each training run taking approximately one day. We train all image and discrete diffusion models on a single node of $4\times$ H100. The training speed (iteration per second) for our image diffusion model is constant across sequence length and label conditioning DiT at about 5.2 it/sec. Training 800 epochs of the image generator takes approximately 10 days. In contrast, our discrete diffusion SEDD training speed scales with the sequence length (see Figure 5). Our large sequence length model, SEDD-medium with L=512, takes about two days to train.

## F License

The compilation of assets used in the reproduction this work is presented in Table 7.

| Asset | License | Source |
|---|---|---|
| ImageNet | imagenet | https://www.image-net.org/ |
| LAION | MIT | https://github.com/LAION-AI/laion-datasets/tree/main |
| DinoV2 | Apache 2.0 | https://github.com/facebookresearch/dinov2 |
| Fast-DiT [Jin, 2025] | CC-BY-NC-4.0 | https://github.com/chuanyangjin/fast-DiT |
| SEDD | MIT | https://github.com/louaaron/Score-Entropy-Discrete-Diffusion |
| LLADA | MIT | https://github.com/ML-GSAI/LLaDA |
| Pytorch 2.5 | Pytorch | https://github.com/pytorch/pytorch/tree/v2.5.1 |
| Transformers | Apache 2.0 | https://github.com/huggingface/transformers |
| This work | MIT | https://github.com/lavoiems/DiscreteLatentCode |

Table 7: Compilation of assets used in the production of this work along with their license and the source location of each asset.

