# OpenReview forum: "Compositional Discrete Latent Code for High Fidelity, Productive Diffusion Models"
_NeurIPS.cc/2025/Conference — NeurIPS 2025 poster_

### Official Review · Reviewer_VouL · 2025-07-02

**Clarity:** 3
**Significance:** 4
**Originality:** 4
**Rating:** 4
**Confidence:** 4

**Summary:**

This paper introduces Discrete Latent Codes (DLC)—a compositional, discrete image representation used to condition diffusion models. DLCs are derived from self-supervised Simplicial Embeddings and represent images as sequences of discrete tokens. The key contributions are: (1) state-of-the-art FID for unconditional ImageNet generation, (2) compositional generation by mixing codes from multiple images, and (3) a text-to-image pipeline where a language model is trained to produce DLCs from text, allowing novel image generation beyond the training distribution.

**Questions:**

1. Why average DLC token embeddings rather than use cross-attention or per-token conditioning? Could more expressive strategies improve performance?
2. How consistent and controllable is compositional generation? Have structured mixing or multi-object compositions been tested? An interesting test would be on part-level generation (e.g., PartCraft [1] / PiT [2]) -- treating part as a discrete species etc.
3. A systematic evaluation on how well does it generalize to unseen prompts or multi-object prompts?
4. How can DLCs generalize to datasets beyond ImageNet, such as fine-grained species, scenes, high-res images, or non-natural data? The discrete latent code can be treated as "concept" too.
5. How would you apply DLCs on existing T2I model (e.g., SD1.5/3 or Flux)?
6. Can we do the reverse on the text-to-DLC -- DLC to text and see how well the image representation can be converted to text prompt (image captioning)?

[1] Ng et al. PartCraft: Crafting Creative Objects by Parts. ECCV 2024.
[2] Richardson et al. Piece it Together: Part-Based Concepting with IP-Priors. arxiv 2025.

**Ethical Concerns:**

["NO or VERY MINOR ethics concerns only"]

**Final Justification:**

I thank the authors for their rebuttal. Most concerns are addressed except the systematic evaluation. Overall the paper is good and I am keeping my score as borderline accept.

**Limitations:**

Yes. The main limitation of using discrete code is the computational tradeoff as discussed in Section 5.

**Quality:**

4

**Strengths And Weaknesses:**

Quality:
- The paper is technically robust and presents a well-executed system that integrates several recent advances (self-supervised vision encoders (SEMs), discrete diffusion models, and DiT).
- SOTA performance on unconditional ImageNet generation, surpassing both unconditional and conditional baselines. The authors run extensive experiments comparing discrete vs. continuous conditioning, ablate different DLC configurations (e.g., token count vs. vocabulary size), and show clear improvements in both fidelity and diversity.
- The compositional generation capability, enabled by simple token-level mixing, is a novel and effective demonstration of the model's expressiveness. Furthermore, the text-to-image results -- though limited in scope -- demonstrate that a pretrained LLM can be repurposed to generate discrete image codes, offering a flexible and low-resource alternative to traditional text-conditioned diffusion pipelines.

Clarity:
- The writing is generally clear, and the paper does a good job tying its contributions back to its stated motivations (i.e., improving fidelity, ease of sampling, and compositionality).

Significance:
- The work presents a meaningful shift in focus: instead of improving diffusion model architectures, it emphasizes improving the conditioning representation. By doing so, it achieves not only higher fidelity but also new generative functionality, such as compositional image synthesis and a bridge between LLMs and image diffusion. The idea of using discrete tokens as a modular interface between modalities (text and image) is compelling and could influence future work in scalable, controllable generation.

Originality:
- The paper presents a novel integration of known techniques to produce a new capability. While discrete latent representations and self-supervised embeddings are not new, using them as conditioning signals for diffusion models, and demonstrating their compositionality via token-level mixing, is a novel and creative contribution.
- Additionally, the use of a language model to generate image tokens directly is an original twist on text-to-image generation, enabling a lightweight and modular alternative to large-scale CLIP-based pipelines.

Weaknesses:
- The pipeline involves multiple stages (encoder training, token inference, discrete diffusion prior training, image diffusion), which demands substantial compute. This may hinder reproducibility and adoption in resource-constrained environments.
- The methodology is presented in a logical progression. However, some technical details could benefit from more intuitive explanation --especially the use of Simplicial Embeddings, the SEDD discrete diffusion prior, and the rationale for averaging DLC token embeddings instead of using more expressive conditioning mechanisms like cross-attention.
- The choice to average token embeddings for conditioning, rather than using sequence-aware mechanisms (e.g. cross-attention), may limit the expressiveness and spatial control of generation.
- The text-to-image pipeline involving LLaDA is quite dense and could be clarified further for readers unfamiliar with diffusion LMs.
Evaluation scope: All experiments are on ImageNet (256×256), with limited exploration of more complex datasets (e.g., multi-object scenes, higher resolutions). It’s unclear how well DLCs generalize beyond the object-centric domain.
- While promising, the text -> DLC -> image pipeline is evaluated on just a few hand-picked prompts. There's no quantitative comparison to standard text-to-image models, nor is it clear how broadly the LLM generalizes to unseen concepts.

---

> ### Author Rebuttal · Authors · 2025-07-30
>
> Thank you for your thorough review and constructive feedback. We are glad you appreciate our contributions and the overall proposed system. We respond to comments and questions below
>
> ## Computation cost of training DLC
> The overall computation is comparable to baselines methods (e.g. RCG, REPA). Moreover, the largest computational cost is the image generator (see paragraph Computational resources). As shown in Figure 4, training the image generator with DLC converges much faster than SOTA baselines. Thus, DLC may enable institutions with less compute to participate in image generation research.
>
> ## Reproducibility and applicability of the work
> To enable reproduction and application of the work, we release every model (the SEM encoder, the DiT, the SEDD and the fine-tuned LLADA) and the ImageNet DLC dataset. Everything is released as a huggingface model or dataset, which makes it straightforward for researcher and practitioner to build on top. We also provide the code for training and evaluating every model and generating the ImageNet-DLC dataset.
>
> ## Using the cross-attention mechanism instead of averaging the token embedding
> This is a great suggestion. We could use a cross-attention mechanism, similar to the one used in SOTA text-to-image models; e.g. stable-diffusion. This mechanism would enable a contextual conditioning at every attention block which would likely improve performance and a more straight-forward application of DLCs to existing T2I. For this work, we argue that simple averaging leads to a closer apple-to-apple comparison to the baseline methods for unconditional generation.
>
> ## On text-to-image evaluation
> While our primary focus is improving image representations for image generation, we also explored text-to-image generation as a proof-of-concept on how such representation may be used in that setup (i.e. as tokens for a LLM). Due to compute and data limitations, we did not aim to outperform existing SOTA text-to-image models. Instead, we aimed to demonstrate the potential of DLC as a bridge between language and vision, and we hope our initial results inspire future work by the broader community.
>
> ## How can DLCs generalize to datasets beyond ImageNet?
> As shown in Figure 6, Figure 7 and Figure 12 in the Appendix, DLCs can generalize to sample images outside of ImageNet. For example, the image generator can generate teapot in Antarctica, despite ImageNet not having any images of teapots in Antarctica. However, as we see in Figure 12, ImageNet has images of teapots and images of snowy mountains (presumably Antarctica). Thus, certain components of the DLC must encode concepts about teapots while others must encode concepts about snowy mountains, as noted by the reviewer. This shows that a SSL objective + an inductive bias for learning discrete representation (i.e. SEM) is enough for learning such concepts. We refer the reviewer to Section 4.4 of [[0](https://arxiv.org/abs/2204.00616)] for an qualitative and quantitative analysis of the SEM representation itself.
>
> ## How would you apply DLCs on existing T2I models?
> We believe that it would be straight-forward to use DLCs in existing T2I models using the cross-attention module proposed by the reviewer. A straightforward approach is to apply the architecture of [[1](https://arxiv.org/abs/2112.10752)] and replace (or add) DLCs conditionally generated from a text prompt as conditioning. Such DLCs may be projected onto a vector of appropriate dimension using a learnable linear projection.
>
> ## Can we do DLC->text
> Yes, the fine-tuned LLM can output text tokens given a DLC. Qualitatively, the outputs are reasonable and we believe that direction is exciting as it enables to combine both image and text tokens within the same generative model.
>
> [0] https://arxiv.org/abs/2204.00616
> [1] https://arxiv.org/abs/2112.10752

---

> ### Comment · Reviewer_VouL · 2025-08-05
>
> I thank the authors for their rebuttal. Most concerns are addressed except the systematic evaluation. Overall the paper is good and I am keeping my score as borderline accept.

---

> > ### Author Response · Authors · 2025-08-05
> > **Thank you**
> >
> > Thank you for your comments.

---

### Official Review · Reviewer_JhZC · 2025-07-02

**Clarity:** 3
**Significance:** 4
**Originality:** 4
**Rating:** 5
**Confidence:** 4

**Summary:**

The authors propose DLC as a novel image representation derived from Simplicial Embeddings, trained with a self-supervised learning objective. DLCs consist of sequences of discrete tokens, contrasting with traditional continuous image embeddings. Diffusion models trained with DLCs achieve enhanced generation fidelity, setting a new state-of-the-art for unconditional image generation on the ImageNet dataset. The compositional nature of DLCs allows for the generation of novel, out-of-distribution images by creatively combining the semantics of different images. The authors also shows the ability to generate image with large LLMs with promising results.

**Questions:**

1. I am confused by the sections "Setup for training image diffusion" and "Setup for latent embeddings diffusion". Can the authors further explain what is the difference between those two and what you are trying to accomplish here?

2. Runtime analysis. I would like to ask the inference time for the proposed method.

Others see weakness

**Ethical Concerns:**

["NO or VERY MINOR ethics concerns only"]

**Final Justification:**

I thank the authors for their rebuttal and I am satisfied with the results. Since I have already given accept, I will not change my score.

**Limitations:**

Yes

**Quality:**

4

**Strengths And Weaknesses:**

Strength:
1. The idea is novel and interesting. I wonder if the discrete latent code could really separate the concepts with a deeper analysis.
2. The experiment results are strong.

Weakness:
1. The images generated by the proposed method (Figure 6 (b)) shows undesired results where the identity of the dog is not preserved (top two images) while the traditional method (d) can adhere to the prompt.
2. I feel like the compositional property is not well explored, does it matters how you concat two DLCs together (interleaving/shuffle/A|B or B|A)? This is interesting explore since you mention the bag of words problem posed by the VLMs.

---

> ### Author Rebuttal · Authors · 2025-07-30
>
> We sincerely thank the reviewer for their detailed and constructive assessment of the work. We are pleased that you find our method novel and impactful, and we appreciate your insightful questions and suggestions. We respond to the comments and questions below
>
> ## Compositionality: How are DLCs combined? (interleaving/shuffle/A|B/B|A)
> The way that DLCs are composed certainly matters as found in Figure 6b. Interleaving, A|B and B|Compositions are possible combinations that could be sampled and would lead to different image production. Since the DLC position is important then shuffling the DLC and then composing them would result in nonsensical image generation.
>
> ## Preservation of identity in generated images:
> Some of the dogs in Figure 6b are not identical to the reference dog in Figure 6a. E.g. The dog in the top right corner has a piece of meat in place of its head and has pasta in place of its fur. We argue that this is a desired composition demonstrating that the image generator can coherently combine features from either reference images. We note that other combinations will more closely preserve the dog identity and this may be forced by sampling more tokens from the source dog image instead of uniformly sampling the tokens as we did in this experiment.
>
> ## Runtime analysis
> The inference compute cost can be controlled by changing the number of sampling steps of either DiT or SEDD. Following the reviewer’s comment, we performed an additional experiment where we control the runtime of images generated with DLC by reducing the number of SEDD steps. We report the comparative results (without CFG) below
> * RCG, runtime = 1.52s/it, FID = 3.44
> * DLC (# SEDD steps = 128), runtime = 1.32 s/it, FID = 4.14
> * DLC (# SEDD steps = 256), runtime = 1.91 s/it, FID = 2.86
> * DLC (# SEDD steps = 1024), runtime = 5.33 s/it, FID = 2.71 (updated result)
> * DLC (# SEDD steps = 2048 + remasking), runtime = 9.89 s/it, FID = 2.0 (updated result)
>
> We find that DLC is competitive with baselines for similar inference runtime but strongly outperform them when allowed for more runtime, showing strong test-time scaling.
>
> ## Clarification: “Setup for training image diffusion” vs “Setup for latent embeddings diffusion”
> These two paragraphs aim at giving the reader more context about the training of the DLC-to-image conditional generative model (DiT) and the training of the DLC generative model (SEDD) respectively. We will revise Section 4.1 to make this distinction clearer, renaming the paragraph headers to “Pixel-space Diffusion Training” and “DLC-space Diffusion Training.”

---

### Official Review · Reviewer_v32v · 2025-07-03

**Clarity:** 3
**Significance:** 3
**Originality:** 3
**Rating:** 4
**Confidence:** 3

**Summary:**

This paper proposes Discrete Latent Codes (DLCs) as a new conditioning representation for diffusion-based image generation. A DLC is a tokenized image representation obtained from a Simplicial Embedding model trained self-supervised, which maps an image into a sequence of discrete tokens instead of a continuous embedding. The authors argue that DLCs are easier to model (since one can learn a discrete token distribution) and inherently compositional, meaning parts of different images can be combined to generate novel outputs.

**Questions:**

Could you elaborate on the limitations and consistency of the DLC compositionality experiments?

**Ethical Concerns:**

["NO or VERY MINOR ethics concerns only"]

**Final Justification:**

The rebuttal has addressed most of my concerns.

**Limitations:**

Yes

**Quality:**

3

**Strengths And Weaknesses:**

**Strengths:** The paper is well-executed with strong experimental results. The proposed DLC conditioning yields **state-of-the-art** unconditional ImageNet generation quality, significantly improving over prior approaches. For instance, a diffusion transformer (DiT-XL/2) conditioned on a 512-token DLC achieves an FID of *~2.2* without guidance, outperforming previous representation-conditioned models (e.g. a recent baseline “RCG” had FID 3.44). With classifier-free guidance, the method further reaches FID 1.47, closing the gap to class-conditioned generators.

**Weaknesses**: In terms of **significance**, the work is strong for unconditional generation research, but one could question how broadly impactful that is, given that *conditional* (especially text-conditional) generation is the main focus of the field now. The paper’s answer is that DLCs can improve those too, but the current evidence for text-to-image is not state-of-the-art (since only 9M pairs were used, the text-conditioned results are likely below the best alternatives).

---

> ### Author Rebuttal · Authors · 2025-07-30
>
> Thank you for your comments. We are encouraged that you found our experimental results strong and have a positive assessment of our paper’s clarity and execution.
>
> ## Significance of this work
>
> Text-to-image is indeed a major focus of the field but all current SOTA models (Midjourney, DALLE, Imagen) are closed source and for us to create a SOTA model would require significant compute, data, and engineering resources which are unavailable to most organizations (including ours).
>
> Our work is impactful since SOTA models may still use basic, continuous representations, e.g. Stable Diffusion 3.5 uses CLIP embeddings. We examine the underexplored direction of discrete image representation and show they are superior to their continuous counterpart for image generative modeling. We believe that SOTA on unconditional generation and DLC’s scaling behaviour in both performance (Figure 5) and efficiency (Figure 4) demonstrates strong promise for improving text-to-image at SOTA scale and compute. For that reason, we believe that this work is of great interest for the NeurIPS community and hope that it will inspire researchers at labs with the appropriate amount of resources to experiment with DLC.
>
> ## Compositionality experiment limitations
> We acknowledge that the compositionality experiments are qualitative. This is consistent with prior work (e.g. [[0](https://proceedings.neurips.cc/paper_files/paper/2024/hash/e304d374c85e385eb217ed4a025b6b63-Abstract-Conference.html
> )]), as we are unaware of standard quantitative benchmarks for compositionality in natural images.
>
> Developing such metrics is non-trivial due to the open-ended nature of visual composition, but is an important direction for future work. Our qualitative results suggest that DLCs enable diverse and coherent recombination of semantic parts, which we believe provides a meaningful step forward.
>
> ## Compositionality experiment consistency
> The same mix of tokens (e.g. AABBA) consistently results in the same outcome (e.g. dog in pasta vs dog made of pasta). Different token mixes result in different semantic mixes and the variability at the semantic level is controlled by the DLC. The variability from the prior of the image generative model captures fine-grained details that are not captured by the DLC such as slight difference in object positioning.
>
> [0] https://proceedings.neurips.cc/paper_files/paper/2024/hash/e304d374c85e385eb217ed4a025b6b63-Abstract-Conference.html

---

> ### Author Response · Authors · 2025-08-05
> **Discussion on significance**
>
> We thank the reviewer for acknowledging and reading our response. We are unclear if our response addressed their concern. In case the reviewer has remaining concerns about the significance of this work, we would like to have a discussion with them to better understand their point of view. To ground the discussion, we refer to the definition of significance from: https://neurips.cc/Conferences/2025/ReviewerGuidelines.
>
> We would also like to note previous similar works as ours published at NeurIPS or similar conferences as precedent of what constitute interesting work for the community:
> - REPA, published at ICLR 2025 as an oral has already 118 citations as of today. This work has inspired large scale works including text-to-image models. Yet, the publication does not have state-of-the-art text-to-image result.
> - RCG, published at NeurIPS 2024. This work also do not have any text-to-image state-of-the-art results.
>
> While we agree with the reviewer that state-of-the-art text-to-image results constitute (very) impactful results, these two works demonstrate that such results are not necessary to constitute impactful work for the community. In fact, the reviewer has already noted in their initial review that our work yields very strong result for image generation as our approach outperforms the two aforementioned published works (i.e. this work address a difficult task in a better way than previous work). We also argue that this work advances our understanding on the topics of modeling diverse data distribution.
>
> With that said, we would like to understand the reviewer's opinion if they still have concern on the significance of this work.
>
> Thank you!

---

### Official Review · Reviewer_KXs6 · 2025-07-03

**Clarity:** 3
**Significance:** 2
**Originality:** 3
**Rating:** 4
**Confidence:** 3

**Summary:**

This paper introduces Discrete Latent Code, which is an image representation derived from simplicial embeddings trained with self-supervised learning and diffusion models. It aims to generate discrete token sequences that are learnable, compositional, and efficient for modeling complex and diverse distributions. The experiments demonstrate that the DLC improves generation quality in unconditional image generation on the ImageNet dataset in terms of FID. Besides, DLCs can do semantically compositional generation by combining tokens from other images. It can also support text-to-image generation by finetuning the large language models. The authors claim that DLC is a representation that bridges the gap between image expressivity and language compositionality.

**Questions:**

1. Could the authors provide visualizations of what individual DLC tokens represent, such as attention maps?
2. Can DLC be used on other datasets or other domains? Can DLC be used in image editing?
3. How does text-to-DLC perform when the prompt is very complex?

**Ethical Concerns:**

["NO or VERY MINOR ethics concerns only"]

**Final Justification:**

The authors have provided detailed responses to my concerns regarding computation cost, applicability to other datasets, and potential application in image editing. However, text-to-image evaluation remains only partially addressed. Therefore, I maintain my original score.

**Limitations:**

Yes

**Quality:**

2

**Strengths And Weaknesses:**

Strengths:
1. The motivation is straightforward. The paper aims to learn an image representation to model the complex data distribution, which also enables image generation and composition to improve the representation. It also addresses limitations of continuous embeddings in generative modeling.
2. The method is original. The paper combines the simplicial embeddings with discrete diffusion in token generation using a unified pipeline.
3. The experiment is clear. The paper provides plenty of experiments. DLC-conditioned diffusion models achieve better performance on ImageNet in terms of FID. Besides, the paper provides extensive ablation studies and compares them with both discrete and continuous baselines.


Weaknesses:
1. The computation cost is large. The method needs to train both DLC and diffusion models, which requires large computing resources. It limits the application and reproduction.
2. Text-to-image evaluation is limited. The text-conditioned DLC model is only evaluated on a small dataset. The human or user studies are also missing.
3. Some analysis about interpretability is missing. The reason why compositional discrete tokens can help generalization in diffusion is unclear. Besides, the semantics of individual DLC tokens are not explored.

---

> ### Author Rebuttal · Authors · 2025-07-30
>
> Thank you for your comments and feedback. We are glad that you appreciated the originality of the method, the clarity of the experiments and the motivation behind the method. Below, we respond to your comments and questions
>
> ## Computation cost of DLC training
> The overall computation is comparable to baselines methods (e.g. RCG, REPA). Moreover, the largest computational cost is the image generator (see paragraph Computational resources). As shown in Figure 4, training the image generator with DLC converges much faster than SOTA baselines. Thus, DLC may enable institutions with less compute to participate in image generation research.
>
> ## Reproducibility and applicability of the work
> To enable reproduction and application of the work, we release every model (the SEM encoder, the DiT, the SEDD and the fine-tuned LLADA) and the ImageNet DLC dataset, all available on huggingface. We also provide the code for training and evaluation of every model and for generating the ImageNet-DLC dataset.
>
> ## Can DLC be used on other datasets or other domains?
> Yes, DLC is not restricted to ImageNet. We used DLC with LAION allowing us to sample images that are not in ImageNet (e.g. “a teapot in Antarctica” is not in ImageNet as discussed in Appendix D, Figure 12). DLC could likely be extended to other domains such as audio, though such exploration is left for future work.
>
> ## Can DLC be used in image editing?
> DLC enables semantic-level image editing through token manipulation, as shown in our compositionality experiments. Unlike pixel-level edits (e.g., inpainting), our method allows for structural changes at the representation level, enabling meaningful semantic transformations.
>
> ## On text-to-image evaluation
> While our primary focus is improving image representations for image generation, we also explored text-to-image generation as a proof-of-concept on how such representation may be used in that setup (i.e. as tokens for a LLM). Due to compute and data limitations, we did not aim to outperform existing SOTA text-to-image models. Instead, we aimed to demonstrate the potential of DLC as a bridge between language and vision, and we hope our initial results inspire future work by the broader community.
>
> ## Interpretability of DLC – visualization of DLC
> We refer the reviewer to Section 4.4 of the SEM paper [[0](https://arxiv.org/pdf/2204.00616)]. The authors demonstrate that SEMs lead to an interpretable, disentangled representation. Our semantic token mixing experiments support this, as meaningful compositions arise from swapping tokens across images. This result indicates that individual tokens carry localized semantically consistent information.
>
> [0] https://arxiv.org/pdf/2204.00616

---

> > ### Comment · Reviewer_KXs6 · 2025-08-05
> >
> > The authors have provided details responses to my concerns, particularly regarding the computation cost, experiments on other datasets, and application on image editing. However, my concern about text-to-image evaluation remains only partially addressed. While the authors acknowledge that their work is not aimed at implementing SOTA text-to-image generation and present their experiments as a PoC, the evaluation lacks sufficient evidence to demonstrate the practical viability of DLC on text-to-image generation tasks. Therefore, I maintain my original score.

---

> > > ### Author Response · Authors · 2025-08-05
> > >
> > > Thank you for your comments. We understand and respect the reviewer's assessment on our work, but want to emphasize the following points.
> > >
> > > While we are confident that our approach for text-to-image would have competitive scaling properties in comparison to existing approach for encoding images, we could not validate that claim due to our limited resources and thus have not made such claim in our paper. Instead, the text-to-image results are introduced as a way to show that the image generator can generate samples that are outside of the image generator training distribution. It was important for us to introduce the text-to-image results, albeit not perfect, because we wanted to encourage labs with the appropriate resources to try out that idea.
> > >
> > > We acknowledge the lack of quantitative measure for our text-to-image results. However, we are not aware of any work that proposed a text-to-image model at our scale. In fact, we believe that obtaining a text-to-image model at our data scale (9M image-caption pairs) is already impressive.

---

### Official Review · Reviewer_RBRg · 2025-07-04

**Clarity:** 3
**Significance:** 3
**Originality:** 2
**Rating:** 4
**Confidence:** 2

**Summary:**

This paper explores the discrete latent representation for conditional/unconditional image generation with diffusion models. Specifically, authors utilize the SEM encoder and argmax operation to extract discrete latent codes (DLC), which are used to condition the diffusion models. The proposed method is compared with various unconditional generative methods using the ImageNet benchmark.

**Questions:**

- For the compositionality of continuous representations, I wonder whether using random position-wise sampling (as done with DLCs) instead of averaging would yield similar results.
- For unconditional generation, since DLCs are sampled with SEDD in a non-autoregressive manner, have you observed quality-diversity trade-offs when varying the sampling temperature or related parameters?

**Ethical Concerns:**

["NO or VERY MINOR ethics concerns only"]

**Final Justification:**

I have read the author's response. While I still have some concerns, such as the smooth semantic interpolation of the DLS tokens due to rebuttal policy, considering the paper's quality and impact on the community, I decided to keep my rating at 4: weak accept.

**Limitations:**

See weaknesses.

**Paper Formatting Concerns:**

No major concerns on paper formatting.

**Quality:**

4

**Strengths And Weaknesses:**

Strenghts

- The paper is well-written, and the motivation for utilizing discrete latent code (DLC) for dense description of images is reasonable.
- Trained DLC can be widely applied to various tasks, including unconditional/text-conditional generation and compositional image synthesis.
- Thoughtful experiments demonstrate the effectiveness and scalability of DLC.

Weaknesses

- While the DLC is promising, experiments are only conducted using the SEM encoder. More detailed analysis on choosing the SSL encoder model, or an ablation study, would strengthen the claims.
- DLCs demonstrate diversity through random position-wise sampling (Figure 6). However, the visual quality of generated samples appears relatively low. Moreover, although the paper emphasizes compositionality, it is unclear whether DLCs support controllable interpolation. I wonder if DLC also shows smooth semantic transfer by adjusting the sampled token ratios as continuous embeddings do via weighted averaging. It would also be helpful to evaluate the diversity-quality trade-off using FID, in addition to the Vendi score, to better assess both aspects jointly.

---

> ### Author Rebuttal · Authors · 2025-07-30
>
> We thank the reviewer for their thoughtful assessment of our work. We are pleased that they appreciated the quality of the paper and the generality of DLC. Below, we respond to their comments and questions
>
> ## Motivation for the SEM encoder.
> Our motivation for using the SEM encoder is informed from Table 2 of [[0](https://arxiv.org/abs/2204.00616)] where discretized SEM achieves a higher linear probe accuracy. Importantly, as shown in Figure 5.a of [[1](https://arxiv.org/abs/2410.06940 )], the downstream linear probe accuracy strongly correlated to the downstream FID. While we agree valuable to provide an ablation over different discrete encoder methods, the cost of running such an experiment is high given our resources.
>
> ## Visual quality evaluation of semantic compositional generation
> To assess visual quality in compositional generation, we reproduce the setup from Table 2 of the paper. Specifically, we compare continuous and discrete representations under equal training budgets and model sizes. We generate 50,000 composed samples using:
> * DLC (token sampling): FID = 34.34
> * DINOv2 (embedding averaging): FID = 41.71
>
> These results indicate that DLC supports more coherent compositional generation. We will include this quantitative result and expand discussion of the diversity-quality tradeoff in the main paper as suggested.
>
> ## Position-wise sampling of continuous representation leads to degenerate generation.
> Qualitatively, the image generated lacks any structure found in natural images and quantitatively we observe a FID of 59.65 in this setting. While position-wise sampling of continuous representation would improve the diversity of the image generation, the continuous embeddings are global and do not allow the same token-wise decomposition of DLC. This aligns with the observation that continuous embeddings lack the localized semantics needed for token-wise decomposition, highlighting a key advantage of DLCs.
>
> ## Continuous interpolation with DLC
> Following the reviewer’s suggestion, we implemented interpolation by adjusting the token sampling ratios between two DLCs, instead of fixing the sampling to be uniform across DLCs. As we interpolate the sampling ratio of the DLCs, we observe meaningful semantic transitions of semantic features from one image to the other. We will include qualitative examples in the appendix.
>
> ## SEDD generation parameters ablation
> As detailed in Appendix A, we study the effect of the resampling ratio in our sampling procedure. We observe that increased resampling improves generation quality while having limited impact on diversity. Our sampling is based on the Tweedie denoising framework [[2](https://arxiv.org/abs/2310.16834); algorithm 2], which does not include a temperature parameter.
>
> * [0] https://arxiv.org/abs/2204.00616
> * [1] https://arxiv.org/abs/2410.06940
> * [2] https://arxiv.org/abs/2310.16834

---

### Comment · Area_Chair_5tW1 · 2025-08-03

Dear Reviewers,

Thanks for your hard work during the review process. We are now in the author-reviewer discussion period.

Please (1) carefully read all other reviews and the author responses; (2) start discussion with authors if you still have concerns as early as possible so that authors could have enough time to response; (3) acknowledge and update your final rating. Your engagement in the period is crucial for ACs to make the final recommendation.

Thanks,

AC

---

> ### Comment · Area_Chair_5tW1 · 2025-08-05
>
> Dear Reviewers,
>
> Please note that submitting mandatory acknowledgement without posting a single sentence to authors in discussions is not permitted. Whether your concerns have been addressed or not, please do tell the authors. Please also note that __non-participating reviewers will receive possible penalties of this year's responsible reviewing initiative and future reviewing invitations.__
>
> Thanks,
>
> AC

---

### Note · Authors · 2025-08-15

We extend our most sincere gratitude towards the reviewers and the ACs that served as member on the reviewing committee of our paper.

Our paper exposes a severe limitation of the current state-of-the art generative models on images. Their struggle to model highly modal distributions has important consequences on the downstream quality of the generation. The solution presented in this paper is simple and well motivated, as pointed out by the reviewers, and results in:
* Strong unconditional generation which does not rely on classifier-free guidance,
* Faster training,
* Localized compositional generation.

We open our paper with a potentially new paradigm for multi-modal generative model which we hope will inspire the members of the AI community and help mitigate some of the outstanding issues of Vision Language Models.

---

### Decision · Program_Chairs · 2025-09-17

**Decision:**

Accept (poster)

**Comment:**

This paper presents a novel method that explores learning discrete latent representations from images in a self-supervised way for conditional/unconditional image generation with diffusion models. Reviewers acknowledged the motivation, method design, and strong performance of the proposed method, while initially raising some concerns, such as high computation cost, unclear generalizability to other datasets, and limited text-to-image generation results.

After the rebuttal, the authors addressed most of the concerns, and all reviewers agreed to accept this paper. AC read all the reviews, author rebuttals, and the paper, and believes this is a good paper for generative modelling and recommends acceptance.